# Economic impact and disease burden of COVID-19 in a tertiary care hospital: A three-year analysis

Mari Kanerva[ID][1]*, Kalle Rautava[2], Tiina Kurvinen[1], Harri Marttila[1], Taru Finnilä[1], Kaisu Rantakokko-Jalava[3], Mikko Pietilä[4], Pirjo Mustonen[4], Mika Kortelainen[5]

1 Infection Control Unit, Turku University Central Hospital, The Wellbeing Services County of Southwest Finland, Turku, Finland, 2 Financial Unit, Turku University Central Hospital, The Wellbeing Services County of Southwest Finland, Turku, Finland, 3 Clinical Microbiology Department, Turku University Central Hospital, The Wellbeing Services County of Southwest Finland, Turku, Finland, 4 Hospital Services Administration, Turku University Central Hospital, The Wellbeing Services County of Southwest Finland, Turku, Finland, 5 Health Economics Unit, University of Turku and Finnish Institute for Health and Welfare, Turku, Finland

* mari.kanerva@varha.fi

## Abstract

### Background

The COVID-19 pandemic has increased morbidity and mortality, along with substantial economic repercussions for healthcare systems and communities worldwide. This study describes the costs incurred by one of the biggest university hospitals in Finland during the first three years of the pandemic, from 2020 to 2022.

### Methods

A retrospective analysis was conducted at a 950-bed tertiary care hospital, encompassing data from 2020 to 2022. Hospitalized COVID-19 cases were identified from an automated surveillance program that integrated microbiological and administrative data. Patient-level data, including hospitalization demographics, vaccination status, and outcomes, were collected. Billing costs, indirect costs, and operational data were obtained to assess hospital costs comprehensively.

### Results

During 2020–2022, 2 555 COVID-19-positive patients were treated at the hospital. Of them, 57% were hospitalized primarily for COVID-19 with the hospital billing costs of 14 492 399 € (median 4 137 €/ patient), and 47% of these costs were due to intensive care. Including ER visits of COVID-19 outpatients, admission screening and isolation costs of COVID-19 positive patients hospitalized for other reasons and indirect expenses, the total costs reached 28 899 298 € over three years. Additionally, the hospital incurred losses of income due to postponed elective surgeries.

**Data availability statement:** The data that support the findings of this study are openly available in Kaggle at https://www.kaggle.com/datasets/marikanerva/covid-19-costs-in-a-tertiary-care-hospital.

**Funding:** The funding was provided by an academic coalition of the University of Turku and Turku University Hospital, specifically through the Microbes and Immunity Research Program (MIRP), which supported the first author with a one-week writing leave (project number 80049). the funder had no role in study design, data collection and analysis, decision to publish, or preparation of the manuscript

**Competing interests:** The authors have declared that no competing interests exist.

## Discussion

The economic burden of COVID-19 at the considered university hospital was substantial. Intensive care costs were a significant driver. This study provides a comprehensive overview of the economic and disease burden of COVID-19 at a tertiary care hospital, highlighting the need for strategic planning and financial readiness to address the costs associated with pandemics.

## Introduction

The COVID-19 pandemic has increased the burden of illness and mortality but has also had an enormous economic impact on both healthcare systems and the community.

The SARS-CoV-2 virus has spread rapidly in Europe and other continents since February 2020, and at the first surge, approximately 10% of infected patients developed pulmonary infection and required hospitalization; among them, the case fatality rate was more than 10% [1].

At the end of 2020, mRNA vaccines became available, and massive population vaccination programs were started [2]. During the first few years, most countries aimed at active surveillance used nucleic acid amplification (NAAT)-based diagnostic testing and contact tracing [3]. All these efforts increased health care costs. In addition, the COVID-19 outbreak cancelled elective operations in hospitals and even temporarily closed down some less critical care pathways in many countries [4,5]. This lead to increased care debt and morbidity for outpatients, causing further outpatient health care usage. The mental burden increased, and medical workers were stressed. Finally, the secondary effects of social distancing, such as the lockdown of international travel to prevent transmission, have had a historical effect on the world economy [6,7].

In Finland, the national COVID-19 prevention strategy was based on testing, isolation, contact tracing and transmission prevention [4,8]. After the whole population was offered three doses of vaccination by March 2022, any preventive societal restrictions, including universal masking recommendations, were largely abandoned [8]. However, at that time, along with the increase in Omicron virus variants, the patient numbers and hospitalizations clearly increased, but the disease was milder [9]. Subsequently, new waves of the epidemic have occurred with novel virus variants.

In Finland, the healthcare system is mostly public, making individuals less vulnerable to direct health-related costs. In 2022, 96% of all hospital care and 77% of all health care were financed by the government [10]. COVID-19 was determined to be a generally hazardous communicable disease in Finland by the Communicable Diseases Act until 30th June 2023. During that time, neither the patients nor their home municipalities paid much from the pocket for COVID-19 care, but the government paid reimbursement for hospitals.

In this study, we aimed to describe the disease burden and costs of the COVID-19 pandemic from a large tertiary care hospital point of view to determine how it affected

the hospital economy during the first three years of the COVID-19 pandemic, 2020–2022. We also wanted to determine the alternative costs that were lost for this infection.

## Materials and Methods

We conducted a retrospective analysis of the costs and effects of the COVID-19 pandemic at Turku University Hospital (TUH), a 950-bed tertiary care hospital in Southwestern Finland. The hospital serves a catchment area with a population of 489 210, constituting 8.8% of the entire country.

### Numbers of COVID-19 patients and their clinical information

COVID-19 patients, i.e., patients with a current positive SARS-CoV-2 test result hospitalized during 2020–2022 were identified from an automated infection surveillance registry (SAI, Hospital Antibiotic and Infection Registry, Neotide, Finland) at the hospital. The registry integrates data from the microbiology laboratories of TUH and administrative records. Data included age, sex, medical specialty, intensive care stay and length of stay. Information on vaccination status and reason for hospitalization, i.e., whether COVID-19 (i.e., pneumonia or fever) was the primary cause for the hospitalization or whether the positive SARS-CoV-2 test result was an asymptomatic co-finding in patients hospitalized for other reasons, had been collected in SAI by routine contact tracing activities by the hospital Infection Control Unit during 2020–2022.

The reason for detecting asymptomatic virus carriers was due to routine admission screening of all patients between December 2021 and October 2022, prompted by the hospital infection control instructions.

### Numbers of SARS-CoV-2 tests

Numbers of SARS-CoV2 tests taken at hospital wards, ER or separate outpatient testing sites were obtained via SAI registry from the hospital laboratory.

All Finnish laboratories report their nucleic acid amplification test (NAAT)-positive COVID-19 cases to the National Infectious Diseases Registry (NIDR). From the public website of NIDR we were able to see the total numbers of tests and positive results in the whole county of Southwestern Finland for the epidemiologic curves.

### Cost perspective

We included costs incurred within the hospital, adopting a hospital perspective. These encompassed direct costs related to COVID-19 patient care or isolation, as well as indirect costs arising from hospital infrastructure adjustments, i.e., increased SARS-CoV2 testing, additional personnel, infection prevention management and administrative work. The costs were not discounted as the COVID-19 pandemic was an acute and simultaneous global phenomenon.

### Hospital costs

Hospital billing costs for patients hospitalized due to COVID-19 were acquired from the hospital administration and were used as a proxy for hospital costs. These costs covered daily care costs, i.e., personnel costs, medications, laboratory and radiology costs. For COVID-19 positive patients primarily hospitalized for other diseases, we included only the estimated minimum COVID-19-related costs, i.e., isolation costs and the SARS-CoV-2 test. For these patients, we estimated isolation costs for the entire hospital stay if it was seven days or less, and for the first seven days if it exceeded this duration, in accordance with hospital guidelines for the length of COVID-19 patient isolation.

We analyzed separately the emergency department (ED) costs for patients who had visited ED due to COVID-19 but then discharged. These patients were recognized in the hospital administrative data by the ICD-10 diagnosis code U07.1. at the ED visit and with no related ward admission.

Other direct or indirect COVID-related costs in the hospital, such as extra personnel, extra cleaning, COVID-19 related IT work and changes in infrastructure, were collected from hospital financial account data.

The hospital administrative data provided information on the potential income losses, i.e., numbers of surgical operations and ward closures. We verified the cause of these losses by discussing with the hospital administrative lead about the strategic decisions that had been made on ward closures.

Extra work due to administrative and medical meetings were assessed from meeting memos at the Infection Control Unit and considered them as opportunity cost.

Data analysis was performed separately for the years 2020, 2021, and 2022, due to differences in the characteristics of the pandemic (i.e., wild-type virus or variants) and its control measures (SARS-CoV2 - testing availability, screening activity, vaccinations and the community lockdown) during these years (S1 Table).

Study permission was obtained from TUH Management (VSSHP/2023/161065). Separate ethical approval was not required for a registry survey according to the Finnish study law (Medical Research Act 488/1999). The medical and administrative data were accessed for research purposes on the 24th July 2023. The patient's COVID-19 –related data and the corresponding hospital billing cost were linked though the personal identity code. After data linkage, these codes were removed from the data and the patients could not be identified.

## Results

### Epidemiologic overview in 2020–2022

In the county of Southwestern Finland, 111 552 COVID-19 cases were detected by the NAAT and reported in the NIDR during 2020–2022, representing 7.7% of the reported cases in the whole country. The daily numbers were the highest in spring 2022 (S1 Fig). The yearly incidence rates of the county during 2020–2022 were 7/1000, 45/1000 and 178/1000, respectively.

TUH microbiology laboratory performed 79% (709 988/ 900 792) of all NAAT specimens in Southwestern Finland with the positivity rate of 11.8% (yearly range 1.9%, 4.8% and 38.5%). This testing cost 54 665 920 €, with the mean cost of one positive finding being 619 €. This included both hospital and outpatient testing. Of all the tests, however, only 10% were conducted on TUH inpatients.

**Hospitalized patients.** During 2020–2022, a total of 2555 (2.3% of all NAAT-positive) COVID-19-positive patients were treated in TUH (yearly 4.2%, 2.1% and 4.3%, respectively). Among all the hospitalized COVID-19-positive patients, 1444/2555 (57%) were treated for COVID-19 as the primary reason for hospitalization (yearly; 92%, 91% and 45% of patients, respectively) (S2 Fig and S2 Table). Of these patients, 182 (13%) needed intensive care (yearly; 9.1%, 22.4%, and 6.8%, respectively) (S3 Table).

During the first two years of the pandemic, the median ages of patients hospitalized due to COVID-19 were 54 and 54.5 years, respectively. In 2022, the patients were older, with a median age of 70. Among the > 70 years old group, 13%, 10% and 5% died during the study years. Among the patients hospitalized due to COVID-19, 4/132 (3.0%), 20/457 (4.4%), and 165/863 (19%) were children (<18 years old) during the three study years.

The favorable effect of COVID-19 vaccination on preventing hospitalization and ICU treatment can be indirectly evaluated by the 2.7 times greater proportion of unvaccinated patients among patients hospitalized due to COVID-19 infection than among those hospitalized with COVID-19 as a co-finding (S2 Table). The respective proportion was 4.7 times greater in ICU patients (S3 Table).

### Hospital costs

**Ward and ICU care of patients hospitalized for COVID-19 infection.** There were a total of 9925 hospital days due to COVID-19 infection (Table 1). The median length of hospital stay was 5 days, varying from 6 in 2020 and 2021–4 in 2022. The proportions of all hospital days spent in the ICU was 17% (1659/9925), with a median of 4 days (Table 2). However, the proportion decreased from 28% in 2020 to 4% in 2022.

The total billing costs of hospitalization for patients with COVID-19 were 14 492 399 €, the median cost being 4 137 €, and the mean cost being 10 149 € per patient (Table 3). The median and mean costs per patient and per day were the lowest in 2022.

Almost half (47%) of the billing costs were due to intensive care (6 748 535 €, 4 068 €/day) (Table 4). This proportion decreased from 69% in 2020 to 22% in 2022. Laboratory and imaging costs were only 4.8% and 1.9%, respectively, of all billing costs (Table 4). The costs of medication (remdesivir, ritonavir-nirmatrelvir or immunological treatments) were included in the billing costs.

Among the different medical specialties, infectious diseases and pulmonology had the highest total costs, 8 382 324 € (1 663 €/day) and 4 586 812 € (1226 €/day), respectively. Other medical specialties covered 1 523 263 € (1 332 €/day) of the costs.

**Costs of COVID-19 patients discharged from the emergency department.** In addition, 2 819 patients visited the emergency department (ED) due to COVID-19 infection between 2020 and 2022 but were discharged. The total ED cost of these patients was 2 409 308 €. During the years 2020, 2021 and 2022, these numbers were 167 587 € (855 €/patient),

**Table 1. Yearly numbers of patients and their hospital days with COVID-19 as the primary reason for tertiary care hospitalization.**

| Year | No of patients | No of hospital days | SD | Mean no of days/pt | Median no of days/pt | Min-max no of days |
|------|----------------|---------------------|------|--------------------|----------------------|--------------------|
| 2020 | 132 | 1 098 | 10,4 | 8,32 | 6 | 1-80 |
| 2021 | 456 | 3 792 | 8,14 | 8,32 | 6 | 1-58 |
| 2022 | 856 | 5 035 | 4,9 | 5,88 | 4 | 1-53 |
| Total | 1 444 | 9 925 | 7,65 | 6,99 | 5 | |

**Table 2. Yearly numbers of patients and days in the ICU due to COVID-19.**

| Year | No of patients | No of ICU days | SD | Mean no of ICU days/pt | Median no of ICU days/pt | Min-max no of days |
|------|----------------|----------------|-------|------------------------|--------------------------|--------------------|
| 2020 | 22 (including 1 child) | 303 | 13,68 | 27 | 8 | 1-43 |
| 2021 | 102 (4 children) | 982 | 7,89 | 9,63 | 7 | 1-45 |
| 2022 | 58 (18 children) | 348 | 5,59 | 5,5 | 4 | 1-26 |
| Total | 182 (23 children) | 1659 | 5,96 | 6,38 | 4 | |

**Table 3. Total billing costs of patients hospitalized due to COVID-19.**

| Year | No patients | Total billing costs | SD | Mean per patient | Median per patient | min–max | Cost per day |
|------|-------------|---------------------|--------|------------------|--------------------|---------|--------------|
| 2020 | 133 | 1 884 960 | 33 754 | 14 173 | 3 868 | 463-214 587 | 1 717 |
| 2021 | 458 | 6 615 149 | 24 595 | 14 444 | 4 695 | 620-212 104 | 1 745 |
| 2022 | 863 | 5 992 290 | 10 756 | 6 944 | 3 771 | 0-140 769 | 1 190 |
| Total | 1 454 | 14 492 399 | 20 126 | 10 149 | 4 137 | 0-214 587 | 1 460 |

**Table 4. Yearly ICU, laboratory and imaging costs of patients hospitalized for COVID-19 and their proportion of the total billing costs.**

| Year | ICU costs | Ward costs | Laboratory costs | Imaging costs |
|------|-----------|------------|------------------|---------------|
| 2020 | 1 304 140 (69% | 487 681 (26%) | 68 656 (4%) | 24 483 (1%) |
| 2021 | 4 097 831 (62%) | 2 130 129 (42%) | 285 290 (4%) | 101 899 (2%) |
| 2022 | 1 346 564 (22%) | 4 141 904 (69%) | 357 688 (6%) | 146 134 (2%) |
| Total | 6 748 535 (47%) | 6 759 719 (47%) | 711 629 (5%) | 272 516 (2%) |

677 428 € (923 €/patient) and 1 564 293 € (828 €/patient), respectively. These visits included the cost of SARS-CoV-2-NAAT.

**Covid-19-related costs of patients hospitalized primarily for other reasons.** There were 1 106 patients (yearly 11, 46 and 1054) hospitalized with COVID-19 for 6 341 days. The number of isolation days for these patients was 5 206. Of these 1 106 patients, 59, 167, 149, 122, 106, 117, 68 and 318 patients were hospitalized for 1, 2, 3, 4, 5, 6, 7 or more than 7 days, respectively. The total COVID-19-associated costs for these patients were estimated to be 424 131 € (Table 5). A total of 95% of these expenses were spent in 2022. The COVID-19 test was included in the laboratory costs only if the diagnosis had not been confirmed at an outpatient clinic before hospital admission.

COVID-19 testing according to the university hospital infection prevention guidelines (i.e., universal screening at admission) incurred extra costs even when the test results were negative. The cost of 64 890 negative test results for hospitalized or discharged patients was 6 172 370 €.

**Other hospital costs of COVID-19 care and prevention of hospital transmission.** According to the specific financial account of the costs of the COVID-19 pandemic in hospitals, there were

7 561 090 € extra costs related to COVID-19, which were not budgeted to be covered by the hospital billing costs and were thus added to total costs. This included extra personnel (doctors, nurses, laboratory workers and administrative personnel) dedicated to COVID-19 work at a cost of 2 120 503 €. There were also costs of doctors, nurses and administrative persons working extra hours, for a total of 855 879 €. Extra IT costs and text messages (COVID-19 test results were sent by SMS) and post services due to COVID-19 cost 419 914 €. Extra cleaning services and renovation of hospital premises for COVID-19 care cost 572 626 €. Other material cost related to COVID-19 care was 2 350 148 €. Other indirect costs, including the purchase or hire of technical equipment on premises, cargo, storage and security related to COVID-19 care were 1 242 030 €.

The total estimated additional costs of the COVID-19 pandemic over 3 years in our tertiary care hospital were thus 28 899 298 €, including care costs of COVID-19 patients 14 492 399 €, ED costs for discharged COVID-19 patients 2 409 308 €, isolation costs of COVID-19 positive patients hospitalized for other reasons 424 131 €, negative screening results 6 172 370 € and other hospital costs 7 561 090 €

**Losses of income and opportunity costs.** According to the interviews of the hospital leaders, no intended ward closures were implemented, but some elective surgeries were cancelled in the hospital during the pandemic. However, at the hospital level, the number of ward days was lower during the pandemic years than during the previous comparative

**Table 5. Estimated isolation costs of the 1106 patients hospitalized with COVID-19 (other diseases as a primary reason for hospitalization).**

|  | no per day | Cost per unit | Cost per day | Total cost during 5206 days and 1106 patients |
|---|---|---|---|---|
| Gloves | 46 | 0,05 € | 2,30 € | 23 048 € |
| FFP 2 or 3 masks | 6 | 3€ | 18,00 € | 93 708 € |
| Surgical masks | 8 | 0,05 € | 0,40 € | 2 082 € |
| Water resistant gowns | 15 | 0,05 € | 0,75 € | 3 905 € |
| Goggles/face shields (reusable) | 4 | 3,00 € | 12,00 € | 62 472 € |
| Time for donning and doffing of PPE | 15 | 3 min (26 €/h) | 19,50 € | 101 517 € |
| Extra time for daily cleaning | 1 | 20 min/day (20,50 €/h) | 6,83 € | 35 557 € |
|  | No of patients | Cost per patient |  |  |
| Extra time and material for discharge cleaning | 1106 | 55 min (20,50 €/h) | 3,99 € | 20 783 € |
| Extra material (bleach) for discharge cleaning | 1106 | 0,29 € | 0,06 € | 321 € |
| COVID-19 screening test | 1106 | 73 € (mean) | (15,51 €) | 80 738 € |
| Total |  |  | 63,83 € (+15,51 €) | 424 131 € |

years 2018–2019. Between 2019 and 2020, the number of ward days decreased by 7%, then increased by 4% from 2020 to 2021 and decreased by 5% again until 2022 (S4 Table). The reasons for these fluctuations were multifactorial, i.e., lack of nursing staff and not directly associated with COVID-19.

Between 2019 and 2020, operative specialities reduced the number of elective operations, leading to a reduction in income of 2 016 539 €. These findings varied between surgical specialties (Table 6). Most reductions took place in April-June 2020, during the first months of the pandemic. The change in the income was affected by both the number and type of operations.

COVID-19 also caused substantial other opportunity costs. Specialists and physicians attended numerous areal and national meetings on steering, vaccination and patient care (at least 3000 working hours as counted on meeting minutes) and spent time in planning new care processes for COVID-19 and guidelines. This work replaced other work including developmental activities and scientific projects which potentially lowered the care quality in the future.

## Discussion

The first three years of the COVID-19 pandemic in Southwestern Finland resulted in approximately 112 000 COVID-19 patients and 2555 patients hospitalized either due to the COVID-19 infection or with COVID-19 as a co-finding. The estimated total cost of COVID-19 patient care and COVID-19-related costs during 2020–2022 in a 950-bed tertiary care hospital were 29 M€. In addition, the losses of income due to postponed operations were approximately two million euros.

The direct hospital mean and median costs of a patient hospitalized at a tertiary care hospital due to COVID-19 between 2020 and 2022 was 10 149 € and 4 137 €, respectively, which were within the range of earlier reports from different continents, although mainly from the first year of the pandemic. These costs varied from the mean of USD 6 557 USD (6 250 €) in a Mexican secondary care hospital [11], to 6 818 I$ (520 €) in a Brazilian tertiary care hospital [12], to 10 196 € in a Spanish tertiary care hospital [13], to 13 476 USD (12 860 €) in Saudi-Arabian tertiary care centers [14] and to the median of 11 695 USD (11 616 €) in academic hospitals and 10 942 USD (10 442 €) in community hospitals in the USA [15]. The differences between countries may partly reflect variations in income brackets, disparities in the intensity of required medical care, standards of care, and heterogeneous cost structures in the published studies. In a review article, the total medical costs of ICU patients with COVID-19 ranged 15 fold as the costs were adjusted into Purchasing Power parity (PPP) 2020 [16].

The greatest disease burden in the hospital was faced in 2022 during the emergence of rapidly spreading Omicron variants and the removal of community restrictions. However, due to good vaccine availability, the disease was milder, and fewer patients required ICU treatment than during the previous years. Of the total numbers during the three years, 60% of patients were treated and 46% of the hospital costs were derived from the year 2022.

**Table 6. Reduction in the number of elective surgeries and the related loss or increase of income between 2019 and 2020.**

|  | Reduction in the number of operations | Reduction in income | Increase in income |
| --- | --- | --- | --- |
| Gastrointestinal surgery | 6,9% | 277 892 € |  |
| Plastic surgery | 10,6% | 79 665 € |  |
| Prosthetic joint surgery | 2,6% | 533 934 € |  |
| Rheuma ortopedics | 28,1% | 284 476 € |  |
| Dental surgery | 14.2% | 141 502 € |  |
| Ear nose and throat surgery | 17% | 1 087 871 € |  |
| General surgery | 12,6% |  | 8 043 € |
| Hand surgery | 8,2% |  | 212 971 € |
| Urology | 3,5% |  | 167 787 € |
| Total loss of income |  | 2 016 539 € |  |

The intensive care cost was a significant cost driver, with a total of 6.7 M€ during the study years. During the first two years of the pandemic, ICU costs constituted two-thirds of hospital billing costs, although the number of ICU days covered only one-fourth of all hospital days. In 2022, the proportions decreased to 22% and 4%, respectively.

The number of patients who visited the hospital ER due to COVID-19 but were discharged during the three years was double the number of patients treated in the hospital during the same period, with a total expenditure of 2.4 M€ during the three years. However, the cost of an ER visit due to COVID-19 was almost half the cost of one day of hospitalization.

The estimated isolation costs of those SARS-CoV2 positive patients who were hospitalized primarily due to other reasons were low, 64 € per day. Of this, material costs were the minority, and two-thirds consisted of work, i.e., cleaning or donning and doffing personal protective equipment.

In our study, the proportion of laboratory and imaging costs of total hospital billing costs was also low. This further highlights the fact that, in terms of hospital costs, personnel and the hospital infra are the most expensive cost drivers. The hospital financial account for COVID-19 costs revealed another 3 M€ cost for extra health care personnel and extra hours. Torres-Toledano et al reported that 95% of the costs of hospital stay were due to infrastructure, personnel and direct medical costs for COVID-19 care [11].

The COVID-19 pandemic has also caused considerable burden in hospitals by replacing other work, e.g., when hospital's lead spent time in organizing meetings and staff were moved to give vaccinations. Many scientific and developmental projects, which cannot be easily assessed financially, had to be postponed.

The presented cost analysis covers COVID-19-related costs incurred at the hospital. Outpatient SARS-CoV-2 NAATs were not included. The latter covered 90% of the entire 55 M€ spent on diagnostic TUH microbiology laboratories. Neither outpatient care costs nor the costs of communal primary care hospitals or vaccination programs for the population were included, as these costs further increase the total costs from the whole health care point of view. The strength of this work is that we were able to describe the costs according to true clinical course of hospitalized COVID-19 patients with the comprehensive surveillance registry data at the hospital. Thus, we were able to exclude care costs unrelated to COVID-19 in patients hospitalized due to other reasons. We were also able to gather patient level costs from hospital billing sources through patient identity code linkage. Similar reports have not been published from other Finnish hospitals. However, showing the costs publicly should increase efforts for prevention in the future pandemics.

Apart from the cancellation of some elective surgeries, there were no known ward closures due to the pandemic at TUH, which is the opposite of many other hospitals in the country. However, cohorting patients and making single-bed room arrangements may have led to closure of the other bed in some two-bed rooms, which we could not detect, however. The number of total hospital days at TUH decreased from 2018 to 2019 but then decreased further until 2020, and this fluctuation may have been a sign of this. The lowest number in 2022 was, however, also due to an increasing shortage of nursing staff closing available beds in several wards.

There are some limitations in the study. First, we describe only hospitals costs, but not the outpatient costs nor the total healthcare costs. We describe the costs only from one hospital point of view and the results are best extrapolated in similar university hospital settings. The abundance and details in economic and clinical data may vary between hospitals and this may affect the variety of cost factors in different hospitals. Thus a uniform computational model should be used to make results fully comparable between hospitals. In addition, as we were retrospectively analyzing the administrative data, we could not inspect whether all sectors in our hospital had reported their COVID-19-related costs uniformly, but we consider this as a minor confounder. Thirdly, the billing costs we used, cover the direct medical costs of examinations, procedures and expensive medications but only a fixed average day fee for the personnel costs. Moreover, billing costs represent hospital charging, and from the hospital point of view, they are also income for the hospital. During COVID-19 pandemic, the government reimbursed 2/3 of the total costs for hospitals and the municipalities paid the rest with the taxpayers' money. Patients were not directly charged for COVID-19 care. And finally, without the COVID-19 pandemic, the hospital would have treated other patients with other diseases instead with the same expenditure. Thus from the

hospital point of view, the COVID-19 pandemic caused mainly societal loss by causing care debt and queues. However, extra expenditures of the COVID-19 pandemic were not left without extra need for savings during the following years in all hospitals in the country. Compared to the previous typical annual number of hospital days (> 250 000 annually) and total costs (781.4 M€ in 2022) at TUH, the annual financial burden of COVID-19 was far less than 5% of the total hospital days and expenses. However, these enormous 29 M€ total costs underline the need for financial preparedness for possible future pandemics. In the broader context, the Finnish medical authorities and government adopted a "test and isolate" strategy in response to the pandemic. Consequently, these costs could be considered unavoidable. A reduction in testing could have saved costs but could have led to a higher number of clinical infections, thereby modulating the cost drivers. In the future, hospital preventive strategies should be carefully redefined based on evidence regarding the transmission routes of the causative agent and the effectiveness of available preventive measures, including masks, vaccinations, and pharmaceuticals. For instance, AI-driven decision support systems could optimize the balance between targeted testing and isolation versus universal masking and symptom-based treatment, ensuring both cost-effectiveness and improved health outcomes.

## Supporting information

**S1 Fig. Weekly numbers of NAAT-positive patients reported in Southwest Finland in the National Infectious Diseases Registry.**
(TIF)

**S2 Fig. Weekly numbers of new COVID-19 hospitalizations between March 2020 and December 2022.**
(TIF)

**S1 Table. Annual characteristics of the COVID-19 pandemic and its control measures in Finland during 2020–2022.**
(DOCX)

**S2 Table. Yearly numbers of patients hospitalized due to COVID-19 and with COVID-19 and the number and proportion of unvaccinated patients.**
(DOCX)

**S3 Table. Yearly numbers of patients in intensive care due to COVID-19 and with COVID-19 and the number and proportion of unvaccinated patients.**
(DOCX)

**S4 Table. Total number of hospital care days on TUH during the comparator years 2018–2019 and during the 2020–2022 COVID-19 pandemic.**
(DOCX)

## Author contributions

**Conceptualization:** Mari Kanerva, Tiina Kurvinen, Mika Kortelainen.

**Data curation:** Kalle Rautava, Harri Marttila, Taru Finnilä.

**Formal analysis:** Mari Kanerva, Kalle Rautava, Tiina Kurvinen.

**Funding acquisition:** Mari Kanerva.

**Supervision:** Mikko Pietilä, Pirjo Mustonen, Mika Kortelainen.

**Writing – original draft:** Mari Kanerva.

**Writing – review & editing:** Kalle Rautava, Tiina Kurvinen, Harri Marttila, Taru Finnilä, Kaisu Rantakokko-Jalava, Mikko Pietilä, Pirjo Mustonen, Mika Kortelainen.

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
