## [Decision Letter · Decision Letter 0]

11 Feb 2025

PONE-D-24-47167Economic Impact and Disease Burden of COVID-19 in a Tertiary Care Hospital: A Three-Year AnalysisPLOS ONE

Dear Dr. Kanerva,

Thank you for submitting your manuscript to PLOS ONE. After careful consideration, we feel that it has merit but does not fully meet PLOS ONE’s publication criteria as it currently stands. Therefore, we invite you to submit a revised version of the manuscript that addresses the points raised during the review process.

**Please read the attached reviews carefully. I would like you to make improvements based on the reviewers' comments, and if there is a part where you feel that you cannot or where it is not possible to accommodate their comments, please explain it with key arguments.**

**Furthermore, please go through the entire text and make the necessary corrections (spelling mistakes, for example).**

We look forward to receiving your revised manuscript.

Kind regards,

Iskra Alexandra Nola

Academic Editor

PLOS ONE

**Journal Requirements:**

Hospital provided fund for a short study leave for the first author.

5. We notice that your supplementary figures are uploaded with the file type 'Figure'. Please amend the file type to 'Supporting Information'. Please ensure that each Supporting Information file has a legend listed in the manuscript after the references list.

Reviewers' comments:

Reviewer's Responses to Questions

**Comments to the Author**

1. Is the manuscript technically sound, and do the data support the conclusions?

Reviewer #1: Partly

Reviewer #2: Partly

2. Has the statistical analysis been performed appropriately and rigorously? 

Reviewer #1: N/A

Reviewer #2: No

3. Have the authors made all data underlying the findings in their manuscript fully available?

Reviewer #1: Yes

Reviewer #2: Yes

4. Is the manuscript presented in an intelligible fashion and written in standard English?

Reviewer #1: Yes

Reviewer #2: No

5. Review Comments to the Author

**Reviewer #1: ** The manuscript requires some revision. The revision is not major but neither would it minor in my view. Thus before acceptance, the authors should respond to the feedback. I think it is worth publishing and is an impritant contribution to the literature.

**Reviewer #2: ** While discussing costs is important, it is equally crucial to address the capacity and capabilities developed during the pandemic. This includes improvements in hospital supplies and the training provided to healthcare providers. A comprehensive analysis should highlight how these enhancements can contribute to better healthcare delivery in the future.

6. PLOS authors have the option to publish the peer review history of their article (what does this mean? ). If published, this will include your full peer review and any attached files.

**Do you want your identity to be public for this peer review?** For information about this choice, including consent withdrawal, please see our Privacy Policy .

Reviewer #1: No

Reviewer #2: **Yes: ** Temesgen Kabeta Chala

---

## [Author Response · Author response to Decision Letter 1]

27 Mar 2025

We have now made corrections according to the Journal Requirement Instructions:

1. The main body text mow meets the requirements and author list has been corrected accord to the instructions. Changes can be seen in the Revised manuscript with trach changes.

2. The grant number 80049 (instead of 30009 which was a mistaken) has been corrected to the Funding information.

3. Funding information has been added to the Cover letter: "Funders had no role in study design, data collection and analysis, decision to publish, or preparation of the manuscript" The financial support covered aónly a short study leave and it was provided by an academic researcher coalition of the hospital and university (MIRP, funding 80049) on the first author's application.

4. All the financial data is publicly available but personalized data that could indentify a person cannot be publicly shared. Institutional regulations require that the individualized data should be analyzed within a designated secure environment (Atolli by Turku University), and even pseudonymized data cannot be transferred outside this platform. However, we have ensured that access to the data can be granted upon a justified request, subject to approval, which we believe aligns with international standards for handling sensitive clinical datasets.

5. Supplementary figures have been apploaded under name "Supplementary material"

6. Supportning information has been taken into account in the revision

7 The reference list has been checked.

Spesific answers to reviewers' comments are included in a separate document. I also list the here as requested:

Answers to Reviewers’ comments

We sincerely thank the reviewers for taking the time to read and provide valuable comments on our manuscript, "Economic Impact and Disease Burden of COVID-19 in a Tertiary Care Hospital: A Three-Year Analysis." We appreciate their insightful feedback and suggestions for improving the text. We made the necessary corrections accordingly and listed them below in the AUTHOR lines.

Reviewer #1

Reviewer: The authors should consider addressing the following:

• It is also important to highlight the perspective that is driving the costing study. By including indirect costs, one would assume that some patient related costs would be included but these are not included. The scope of the costs covered and the justification for the choice of perspective should be included in the study.

AUTHOR: Thank you for the comment. The primary perspective of this study is the hospital’s financial standpoint, aiming to quantify the economic burden of the COVID-19 pandemic on a tertiary care hospital and assess its implications for future pandemic preparedness. Consequently, we included both direct and indirect costs that could be clearly attributed to the COVID-19 surge, ensuring that our analysis captures the minimum financial impact while avoiding overestimation.

To clarify this further, we have explicitly stated the cost perspective in the Introduction and Methods sections:

“Introduction: In this study, we aimed to describe the disease burden and costs of the COVID-19 pandemic from a large tertiary care hospital point of view.”

“Methods: Cost perspective

We included costs incurred within the hospital, adopting a hospital perspective. These encompassed direct costs related to COVID-19 patient care or isolation, as well as indirect costs arising from hospital infrastructure adjustments, i.e. increased SARS-CoV2 testing, additional personnel, infection prevention management and administrative work.”

Reviewer: There was no mention of whether costs were discounted and what rate of discounting was used. In the event that authors took a deliberate choice not to discount, there is need for the authors to justify why.

AUTHOR: We thank you for the comment. The costs were not discounted, as the COVID-19 pandemic was a simultaneous and acute global phenomenon, where the economic burden was incurred over a short period without significant time delays in expenditure. Discounting is generally applied to long-term cost assessments where future costs and benefits need to be adjusted to present value. However, in this case, the costs were realized within a three-year window, and discounting would not significantly impact the interpretation of results. Additionally, maintaining undiscounted values ensures comparability with other pandemic-related economic analyses, many of which report direct, nominal costs.

This is now mentioned in the Methods section:

“The costs were not discounted as the COVID-19 pandemic was an acute and simultaneous global phenomenon.”

Reviewer:

• The presentation of the economic burden of COVID 19 cases is critical for hospitals to prepare adequately for future scenarios. However, COVID 19 varied in severity and the costs associated with each severity profile varied markedly, it would be important to stratify the costs by severity profile.

AUTHOR: Thank you for the comment. In this study, we partially accounted for disease severity on costs by analyzing expenses associated with intensive care. As shown in Table 2, a total of 182 patients required ICU care, with a median ICU stay of 4 days at a cost of €4,068 per day. Consequently, the total additional cost per hospital stay for severely ill ICU patients amounted to €16,272 compared to less severely ill patients.

Unfortunately, due to data constraints and the absence of a standardized severity classification system in our dataset, we were unable to apply a more detailed cost stratification across different severity levels (e.g., mild, moderate, severe non-ICU cases). However, we acknowledge the importance of this approach and suggest that future research should incorporate standardized severity scoring systems to enhance cost stratification and preparedness planning.

Reviewer:

It is also important to highlight whether these include costs of long-term effects of COVID 19. I suggest that the description of what a COVID 19 patient is made clearer highlighting the eligibility and inclusion criteria for who was considered a COVID 19 case.

AUTHOR: Thank you for the comment. This study focused exclusively on the acute hospitalization costs of patients with a current positive SARS-CoV-2 test result and did not account for long-term effects or post-COVID-19 complications, such as prolonged rehabilitation, follow-up visits, or chronic conditions related to COVID-19.

To provide a clearer definition of a COVID-19 patient and the inclusion criteria, we have now explicitly stated this in the Materials and Methods section:

“COVID-19 patients i.e. patients with a current positive SARS-CoV-2 test result hospitalized during 2020-2022 were identified from an automated infection surveillance registry”

and

“…whether COVID-19 (i.e., pneumonia or fever) was the primary cause for the hospitalization or whether the positive SARS-CoV-2 test result was an asymptomatic co-finding in patients hospitalized for other reasons…”

Reviewer:

• Kindly indicate medication costs for those who came in ED and those who were hospitalized for other reasons and then got COVID 19.

AUTHOR: Thank you for the comment. The costs of medication (including COVID-19 drugs remdesivir, ritonavir-nirmatrelvir or immunological treatments) were included in the hospital billing costs. However, due to limitations in the administrative data structure, we were unable to disaggregate medication costs specifically for patients who visited the ED or those hospitalized primarily for other reasons who subsequently tested positive for COVID-19.This is mentioned in the Results section:

“The costs of medication (remdesivir, ritonavir-nirmatrelvir or immunological treatments) were included in the billing costs.”

Discussion section:

• Lines 274-280: Kindly present equivalent costs for the currencies that are non-Euro costs. Furthermore, the discussion should be presented better. The costs are similar to the hospital in Spain and somewhat similar to that in USA. These countries are of similar income profile to Finland. While those with lower costs Mexico and Brazil are in a different income bracket. Could this be an explanation? Kindly discuss a bit further

AUTHOR: Thank you for this comment. To enhance comparability, we have now converted all non-Euro costs into Euros. The differences may be due to income brackets as the reviewer suggests. This has been added in the Discussion:

“The differences between countries may partly reflect variations in income brackets, disparities in the intensity of required medical care, standards of care and heterogeneous cost structures in the published studies”

Reviewer

• Lines 281-285: How do these findings compare to findings from other hospitals in Finland, Europe or other continents? Do they confirm what you have found? If not, why not?

AUTHOR: Thank you for your comment. To our knowledge, COVID-19-related hospital cost analyses have not been published from other Finnish hospitals limiting direct national comparisons, whereas there are more recent publications from Europe and other continents. We added a new reference of a review article (16)

“In a review article, the total medical costs of ICU patients with COVID-19 ranged 15 fold as the costs were adjusted into Purchasing Power parity (PPP) 2020 (16).”

Reviewer

• Lines 281-309: This section is restating what has been described in the results section. The authors should discuss how this compares with findings from Finland or other settings and what the implications of their findings are and how they contribute to the evidence-base for hospital and government efforts on pandemic preparedness and response going forward.

AUTHOR:

Thank you for this insightful comment. We acknowledge the need for a broader discussion comparing our findings with studies from other settings and expanding on the policy implications. To address this, we have revised the Discussion section as follows:

“Similar reports have not been published from other Finnish hospitals. However, showing the costs publicly should increase efforts for prevention in the future pandemics.”

… and

“In the broader context, the Finnish medical authorities and government adopted a “test and isolate” strategy in response to the pandemic. Consequently, these costs could be considered unavoidable. A reduction in testing could have saved costs but could have led to a higher number of clinical infections, thereby modulating the cost drivers.”

Reviewer

• Kindly also indicate some future research areas that your research highlights.

AUTHOR: Thank you for your comment. Building upon our findings, we have identified several key areas for future research to enhance pandemic preparedness and response, including evaluation of preventive strategies and studying the cost-effectiveness of different intervention strategies.

We added into Discussion as follows:

“In the future, hospital preventive strategies should be carefully redefined based on evidence regarding the transmission routes of the causative agent and the effectiveness of available preventive measures, including masks, vaccinations, and pharmaceuticals. For instance, AI-driven decision support systems could optimize the balance between targeted testing and isolation versus universal masking and symptom-based treatment, ensuring both cost-effectiveness and improved health outcomes.”

Reviewer #2

Comments

1. Reviewer: There were WASH activities as prevention of infection in most health facilities, while primarily focused for COVID-19 pandemic, they have great value in preventing other infections. At the same time, what are the expected gains that has to be deducted from the costs?

AUTHOR: Thank you for the comment. We acknowledge that WASH activities, including improved hand hygiene and strengthened infection control protocols, were key interventions during the COVID-19 pandemic. We added a sentence about this into the Discussion:

“In the future, hospital preventive strategies should be carefully redefined based on evidence regarding the transmission routes of the causative agent and the effectiveness of available preventive measures, including masks, vaccinations, and pharmaceuticals. For instance, AI-driven decision support systems could optimize the balance between targeted testing and isolation versus universal masking and symptom-based treatment, ensuring both cost-effectiveness and improved health outcomes.”

2. Reviewer: Although the report mentions lost income due to cancelled elective surgeries, it could benefit from a more thorough analysis of the long-term implications of these cancellations on patient care and hospital finances.

AUTHOR: Thank you for the comment. Unfortunately, we are unable to reliably address this intriguing question or determine at this stage which specific care queues can be attributed to the COVID-19 pandemic. Since 2023, the shortage of nursing staff and reductions in healthcare reimbursements in Finland have had a significant impact on the healthcare system. These factors have recently contributed to longer surgery wait times.

This has been mentioned in the Discussion:

“The lowest number in 2022 was, however, also due to an increasing shortage of nursing staff closing available beds in several wards.”

3. Reviewer. The findings would be strengthened by more rigorous statistical analyses to assess the significance of observed trends and differences between groups.

AUTHOR: Thank you for the good suggestion to improve the analyses. However, the data and our descriptive study design do not allow further comparison of patient groups and their clinical or financial characteristics. Nonetheless, we provide a separate description of the costs associated with intensive care, as well as the care costs for patients who visited the emergency room but were not hospitalized.

4. Reviewer: While the study briefly mentions limitations, a more comprehensive discussion of potential confounding factors and the generalizability of the results would enhance the credibility of the findings.

AUTHOR: Thank you for the good comment. We added a few sentences into the Discussion about the confounding and generalizability.

“The abundance and details in economic and clinical data may vary between hospitals and this may affect the variety of cost factors in different hospitals. Thus a uniform computational model should be used to make results fully comparable between hospitals. In addition, as we were retrospectively analyzing the administrative data, we could not inspect whether all sectors in our hospital had reported their COVID-19-related costs uniformly, but we consider this as a minor confounder.”

5. Reviewer: The exclusion of outpatient SARS-CoV-2 NAAT costs and communal primary care costs limits the study's scope, underestimating the total economic impact of the pandemic on the healthcare system.

AUTHOR: Thank you for the comment. We agree with the reviewer that excluding these costs underestimate the total economic impact on the healthcare system. However, including total healthcare costs or even societal costs would have demanded a different study design, additional data sources, and a broader methodological approach, which were beyond the scope of our current hospital-based analysis. We added the following sentence in the limitations in the Discussion.

---

## [Editor Report · Decision Letter 1]

4 Apr 2025

Economic Impact and Disease Burden of COVID-19 in a Tertiary Care Hospital: A Three-Year Analysis

PONE-D-24-47167R1

Dear Dr. Kanerva,

We’re pleased to inform you that your manuscript has been judged scientifically suitable for publication and will be formally accepted for publication once it meets all outstanding technical requirements.

Kind regards,

Iskra Alexandra Nola

Academic Editor

PLOS ONE

Additional Editor Comments (optional):

Dear Authors,

Please add this sentence (from your responses to reviewers) into the paper (maybe into the section about study limitations?):

"However, we acknowledge the importance of this approach and suggest that future research should incorporate standardized severity scoring systems to enhance cost stratification and preparedness planning."

Thank you,

Kind regards,

Iskra A. Nola
---

## [Editor Report · Acceptance letter]

PONE-D-24-47167R1

PLOS ONE

Dear Dr. Kanerva,

I'm pleased to inform you that your manuscript has been deemed suitable for publication in PLOS ONE. Congratulations! Your manuscript is now being handed over to our production team.

Kind regards,

on behalf of

Dr. Iskra Alexandra Nola

Academic Editor

PLOS ONE